# Systematic Evaluation of LoRA Adapter Placement and Rank Allocation for Resource-Constrained Fine-Tuning

## Abstract

Parameter-efficient fine-tuning (PEFT) methods like Low-Rank Adaptation (LoRA) enable adapting large language models (LLMs) under tight computational and data constraints. However, optimal adapter placement and rank allocation remain underexplored in ultra-low-resource regimes. Here we systematically evaluate LoRA configurations-attention-only, MLP-only, and combined placements with ranks 8 and 16-on the MBPP-mini code generation and GSM8K-mini math reasoning datasets using the Qwen2.5-0.5B-Instruct model with fixed training budgets. We provide an initial cross-architecture comparison using Llama-3.2-1B-Instruct. Our hardened evaluation protocol includes deterministic prompting, AST-based code extraction, and import-safe execution. The results reveal task-dependent placement preferences: attention-only placement achieves higher performance than combined placement for code generation (Qwen: 24.0% vs. 19.0% ± 2.2%; Llama: 28.0% vs. 22.5% ± 0.7%), while combined placement shows strong performance for mathematical reasoning (Llama: 48.5% ± 0.7%). While rank scaling achieves comparable performance to increased training steps for parameter efficiency, hardware profiling reveals all configurations fit within 2.2GB GPU memory, enabling consumer-grade fine-tuning. These findings, supported by experiments on two model architectures, provide practitioners with empirically grounded guidance for LoRA configuration selection in resource-constrained settings.

## 1 Introduction

Fine-tuning Large Language Models is computationally expensive (Hu et al., 2022). Even compact models (0.5B-3B parameters) need > 1000 labeled examples and multi-GPU hours to deliver stable results (Dettmers et al., 2023; Liu et al., 2022; Zhou et al., 2024). The cost and availability barriers to such hardware limit researchers, small teams, and practitioners from fine-tuning their models (Dettmers et al., 2023). It motivates the need to democratize low-budget adaptation of low-resource datasets (< 200 examples) accessible on consumer-grade hardware.

Ultra-low-data fine-tuning (< 200 examples) is practically relevant in settings where labeled examples are expensive or restricted: expert annotation for domains such as medical coding, legal analysis, or scientific tagging; rapid feasibility testing before a larger collection campaign; privacy-constrained on-premise adaptation using only a small sample of sensitive data; and personalization

from limited user interaction history. In these scenarios, configuration choices must extract as much signal as possible from minimal supervision.

While LoRA (Hu et al., 2022) enables Parameter Efficient Fine Tuning (PEFT), reducing trainable parameters, adapter placement, and rank scaling (attention vs. MLP) remain underexplored design choices in small datasets ($< 200$ examples) where the focus on increasing the steps dominated the rank scaling (Liu et al., 2022; Zhou et al., 2024).

**Research Questions:** This work is a systematic empirical study of parameter-efficient fine-tuning under resource constraints. We address the following questions:

- **RQ1:** How does LoRA adapter placement (attention-only, MLP-only, combined) affect task performance and computational cost under fixed fine-tuning budgets?

- **RQ2:** How does rank scaling compare to increasing training steps when the total trainable-parameter budget is controlled?

- **RQ3:** Are the observed trends consistent across code generation (MBPP-mini) and mathematical reasoning (GSM8K-mini) tasks, and stable across random seeds?

**Contributions:** Our contributions are primarily empirical:

1. We present a controlled empirical evaluation of LoRA adapter placement strategies (attention-only, MLP-only, combined) under ultra-low-resource settings with fixed token budgets ($<200$ examples).

2. We compare rank scaling versus step scaling under matched trainable-parameter budgets, demonstrating that a higher rank with fewer steps often achieves comparable performance to a lower rank with more steps.

3. We introduce a hardened evaluation protocol for code generation that distinguishes functional correctness from syntactic validity, revealing that naive string-matching inflates reported performance by up to 25% relative.

4. We provide systematic hardware profiling linking each LoRA configuration to concrete resource requirements (GPU memory, training time), providing empirical guidance for feasibility assessment prior to training.

5. We provide initial cross-architecture validation by replicating key experiments on Llama-3.2-1B-Instruct, showing that the task-dependent placement pattern holds across two different model families.

6. We report results with mean $\pm$ standard deviation for key configurations where multiple seeds were run; single-seed point estimates are clearly marked. We provide anonymized supplementary artifacts for reproducing the main results.

## 2 Related Work

### 2.1 Parameter-Efficient Fine-Tuning (PEFT)

Parameter-efficient fine-tuning (PEFT) includes methods that adapt large pretrained models by updating a small set of parameters while freezing the base weights at their original precision (Hu et al., 2022; Houlsby et al., 2019). Earlier methods for fine-tuning included prompt tuning (Lester et al., 2021), which learns prompt embeddings, and prefix-tuning (Li & Liang, 2021), which prepends trainable vectors to the model's attention layer as keys and values. Apart from prompt tuning, we also used adapter modules that added less than 1% trainable parameters (Houlsby et al., 2019; Pfeiffer et al., 2020) as lightweight layers after the attention and feed-forward sublayers. After these methods, we had LoRA, which enabled PEFT fine-tuning by injecting trainable low-rank decomposition matrices into pre-existing weight matrices to control the effective learning rate (Hu et al., 2022). After LoRA, we saw significant changes leading to QLoRA (Dettmers et al., 2023), combining LoRA with 4-bit NormalFloat quantization and paged optimizers. QLoRA enabled 65B-scale tuning on a single GPU by quantizing base weights while keeping LoRA adapters at higher precision (Dettmers et al., 2023). Other variants, such as Infused Adapter by Inhibiting and Amplifying Inner Activations ($\textbf{IA}^3$) (Liu et al., 2022), inject three learnable vectors per layer to rescale activations without adding new layers. In contrast, Weight-Decomposed Low-Rank Adaptation (DoRA) (Liu et al., 2024), which extends LoRA, decomposes the weight into direction and magnitude. Despite progress in various methods, the placement of ranks and adapters remains underexplored (Liu et al., 2022; Zhou et al., 2024).

### 2.2 Low-Rank Adaptation (LoRA)

Low-Rank Adaptation (LoRA) (Hu et al., 2022) enables parameter-efficient fine-tuning by injecting trainable low-rank matrices into pre-existing weight matrices of a frozen transformer. For a weight matrix $W_0 \in \mathbb{R}^{d \times k}$, LoRA models the update as $\Delta W = BA$, where $B \in \mathbb{R}^{d \times r}$ and $A \in \mathbb{R}^{r \times k}$ are initialized with random Gaussian and zero values, respectively, and $r \ll \min(d, k)$ is the rank. During the forward pass, the output is computed as

$$W_0 x + \frac{\alpha}{r}(BAx),$$

where $\alpha$ is a scaling hyperparameter that controls the magnitude of the update independently of $r$. This design ensures that increasing $r$ adds capacity without changing the learning dynamics when $\alpha$ is fixed. LoRA is typically applied to attention projection matrices ($W_q, W_k, W_v, W_o$) although placement remains configurable (Zhou et al., 2024). QLoRA (Dettmers et al., 2023) extends LoRA by quantizing base model weights to 4-bit precision of $W_0$, reducing memory by 60–75% compared to LoRA while preserving adapter precision. In practice, $r \in \{4, 8, 16, 32\}$ and $\alpha \approx 2r$ are common heuristics (Hu et al., 2022).

**Automatic LoRA Configuration:** Several recent works address automatic adapter placement and rank allocation. AdaLoRA (Zhang et al., 2023) dynamically adjusts rank during training by pruning singular values based on importance scores, achieving adaptive rank allocation without manual tuning. We also have AutoLoRA (Zhang et al., 2024), which uses meta-learning to select which layers to apply LoRA adapters automatically. Third is LoRA-drop (Zhou et al., 2025) that proposes dropping redundant adapters during training to reduce computation while maintaining

performance. These methods focus on automating configuration selection during training, often requiring additional computation for importance estimation or meta-learning.

Our work differs in focus and scope: rather than proposing new adaptive algorithms, we provide a systematic empirical study of fixed LoRA configurations under ultra-low-resource constraints ($<200$ examples). Our goal is to derive practical guidance for practitioners who must select configurations before training begins, particularly in settings where the computational overhead of adaptive methods may not be justified. Our findings complement these adaptive approaches by providing informed starting points when adaptive search is infeasible.

### 2.3 Adapter Placement Strategies

Adapters in PEFT are small, trainable modules that update only specific parts of the model, leaving the majority unchanged. The effectiveness of adapters depends on their placement-whether in attention layers, MLP layers, or both, which influences output quality and parameter efficiency. Strategic placement can improve performance without unnecessary memory or compute overhead.

Prior to LoRA, Houlsby-style adapters (Houlsby et al., 2019) inserted "bottleneck modules" after both the attention and feed-forward (MLP) sub-layers of the BERT-base model, updating 3.6% of the parameters by compressing the data to a lower dimension. LoRA (Hu et al., 2022) improved the efficiency by updating low-rank matrices to specific weights instead of "bottleneck layers". LoRA targets the attention "projection" matrices $(W_q, W_k, W_v, W_o)$ to handle how the module computes attention scores and combines results (Zhou et al., 2024). Configuring attention is crucial, as most relational knowledge resides in the model's attention, making it impactful for adaptation with minimal parameters (Diao & Loynd, 2023; Ranjan et al., 2025). However, no prior work systematically compares attention-only, MLP-only, and combined placements under ultra-low-data regimes ($< 200$ examples) with fixed token budgets. In this paper, we run controlled ablations on MBPP-mini and GSM8K-mini datasets, revealing task-dependent placement preferences: attention-only placement outperforms combined for code generation, while combined placement benefits mathematical reasoning.

Fine-tuning a model in low-resource settings ($< 200$ examples) introduces challenges such as overfitting, poor generalization, and optimizer instability. With limited data, models are prone to memorizing the training set rather than learning generalizable patterns, leading to poor performance on unseen data despite high training accuracy. Regularization techniques such as weight decay, dropout, and early stopping can help mitigate these issues. However, their effectiveness diminishes when the number of trainable parameters exceeds the number of examples (Zhang et al., 2021). Data augmentation methods, such as shuffling sample order or rewording prompts, can help build a more effective dataset, but are limited by potential mismatches in data distribution and label consistency.

PEFT approaches, such as LoRA (Hu et al., 2022) and traditional adapters (Houlsby et al., 2019; Pfeiffer et al., 2020), mitigate these issues by minimizing the number of parameters updated, thereby reducing overfitting. However, even in PEFT, deciding on the adapter location (attention vs MLP) and rank remains pivotal. A substantial portion of the few-shot parameter-efficient fine-tuning (PEFT) literature continues to report results on high-resource NLU suites, most prominently GLUE and SuperGLUE, rather than in settings with only a handful of labeled examples per task. For instance, Ahead-of-Time P-Tuning (Gavrilov, 2023) evaluates exclusively on GLUE and Super-

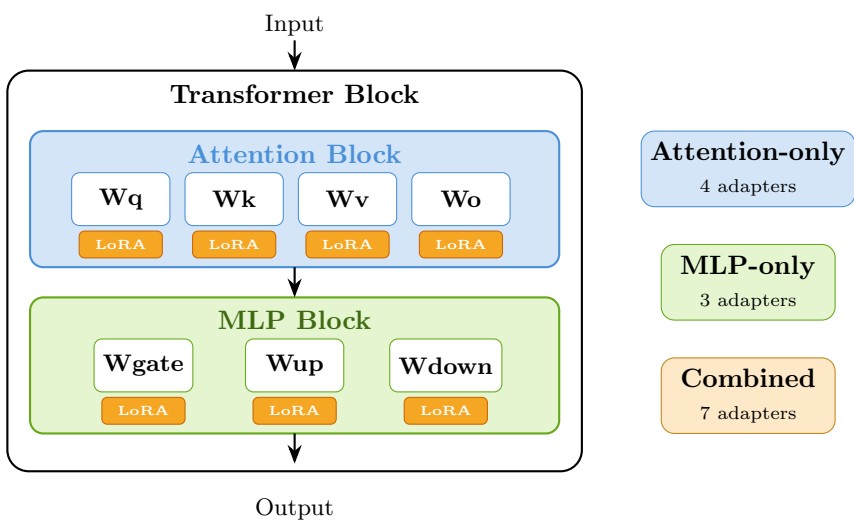

Figure 1: LoRA adapter placement options in a Transformer block. **Attention-only** targets query, key, value, and output projections $(W_q, W_k, W_v, W_o)$. **MLP-only** targets gate, up, and down projections $(W_{gate}, W_{up}, W_{down})$. **Combined** applies adapters to all seven modules. Our experiments reveal task-dependent placement preferences: attention-only outperforms combined for code generation, while combined shows advantages for mathematical reasoning.

GLUE, and P-Tuning v2 (Liu et al., 2021) likewise benchmarks on SuperGLUE, exemplifying the community's reliance on comparatively well-resourced datasets for empirical validation. While such evaluations are informative, they do not target the "true few-shot" regime. As CLUES (Mukherjee et al., 2021) argues, mainstream benchmarks such as GLUE/SuperGLUE give models access to relatively large amounts of labeled data, leaving very small-data ($< 200$ examples) settings comparatively understudied. Prior work explores LoRA in few-shot instruction settings but relies on synthetic data, thereby overlooking computational challenges. Therefore, prior work has explored only limited token budgets while adjusting the adapter location and rank in real-world laptop-scale tuning settings.

## 2.4 Code Generation Benchmarks and Evaluation

Code generation tasks evaluate a model's ability to generate correct, functional programs from natural-language descriptions. The Mostly Basic Python Problems (MBPP) benchmark (Austin et al., 2021b) consists of 974 crowd-sourced Python problems focused on loops, functions, and if-else statements, each with sample solutions and automated tests. For our study, we use a subset of MBPP with 200 training and 100 test samples, as it is well-suited for evaluating performance in ultra-low-resource settings. In contrast, the HumanEval dataset (Chen et al., 2021) contains 164 more complex algorithmic tasks, making it less suitable for experiments with very limited data.

We evaluate code-generation performance using pass@k, the HumanEval metric introduced by Chen et al. (Chen et al., 2021), which estimates the probability that a task is solved when up to $k$ independent samples are attempted; consequently, in one-shot or consistency-focused settings, we report pass@1 as the primary score. Following Chen et al. (Chen et al., 2021), our protocol is execution-based: model completions are run against unit tests to determine functional correctness, using the HumanEval harness. Because executing untrusted code can yield inconsistent outcomes, we adopt the EvalPlus safeguards of Liu et al. (Liu et al., 2023), tight timeouts, I/O suppression, and syscall/subprocess restrictions, together with expanded test suites (HumanEval+) to reduce flakiness and prevent risky behaviors. To make sure that the assessments are reliable in a low-data tuning setting, we introduce a hardened evaluation protocol:

- Consistent prompts enforcing fixed-function names (like def solution(..):)
- AST-based code extraction method to remove the unnecessary add-ons like notes, comments, or explanations
- Import-safe execution in a locked space blocking the modules like os, sys, and sub-processes.

We also track execution success rates alongside pass@1 to identify issues such as syntax errors, import errors, runtime errors, or logic failures.

## 2.5 Mathematical Reasoning and Cross-Domain Transfer

Mathematical reasoning tasks assess a model's ability to perform step-by-step calculations from natural-language word problems. The GSM8K dataset (Cobbe et al., 2021b) contains 8,500 grade-8 level math problems, each accompanied by detailed step-by-step solutions. For our low-resource experiments, we use a subset of GSM8K comprising 200 training and 100 test samples, aligning with our code-generation experiments and enabling consistent evaluation of fine-tuned models.

For evaluation, we use Exact Match (EM) on the final numerical answers, which requires both the correct result and proper formatting. Previous PEFT research on the GSM8K dataset has typically used over 1000 samples (Liu et al., 2022), focusing on prompt engineering or full model fine-tuning rather than systematically exploring adapter placement or rank adjustment under limited compute resources.

### 2.6 Hardware-Aware PEFT Configuration

Existing PEFT research focuses primarily on task accuracy, with hardware considerations treated as secondary constraints rather than first-class design variables. Practitioners typically select LoRA configurations through trial-and-error, risking out-of-memory failures or suboptimal resource utilization (Zhang et al., 2024). While QLoRA (Dettmers et al., 2023) addresses memory efficiency through quantization, and AdaLoRA (Zhang et al., 2023) dynamically adjusts rank during training, no prior work provides upfront profiling that maps configuration choices (placement, rank) to concrete hardware requirements (GPU memory, training time) before training begins.

This gap is particularly acute in ultra-low-resource settings, where failed training runs due to memory constraints represent significant wasted effort. Our work addresses this by systematically profiling all configurations, enabling practitioners to select feasible configurations that perform well under our tested settings and promising configurations under their hardware constraints without exhaustive search.

**Hardware-Aware PEFT Configuration System**

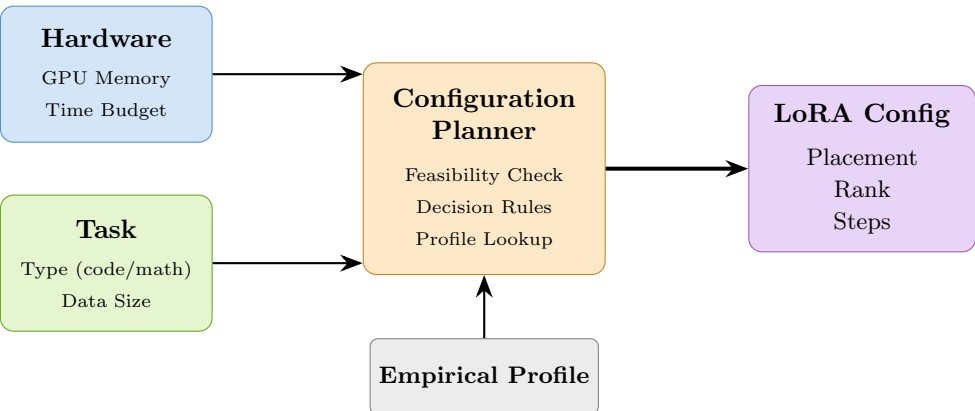

Figure 2: Hardware-aware PEFT configuration system. The planner takes hardware constraints (GPU memory, time budget) and task characteristics (type, data size) as input, consults the empirical performance profile (Table 2), and outputs a recommended LoRA configuration before training begins. It enables practitioners to select feasible configurations without trial-and-error.

## 3 Methodology

### 3.1 Experimental Setup

We conduct all experiments using **Qwen2.5-0.5B-Instruct** (Qwen Team, 2024), a 0.5 billion parameter instruction-tuned language model. We have selected this model for three primary reasons:

  i) performs well on the coding and math tasks despite its compact size.

 ii) it is compatible with consumer GPUs (such as the T4 with 16GB RAM using `bfloat16`), enabling laptop-scale fine-tuning.

iii) its weights are open and use flexible licensing that allows anyone to reproduce our work efficiently.

### Datasets

To compare across various areas, we test on two small benchmarks with the same 200 training and 100 test samples.

- MBPP-mini (Austin et al., 2021b;a): a 200/100 train/test subset of the Mostly Basic Python Problems benchmark covering basic control flow, loops, and if-else logic.

- GSM8K-mini (Cobbe et al., 2021b;a; Contributors, 2024): a 200/100 train/test subset of GSM8K containing grade-school math problems requiring multi-step arithmetic.

MBPP-mini was created by shuffling the full MBPP training split with fixed seed 42, selecting the first 300 examples, and splitting them into 200 training and 100 test examples; we verified zero overlap using exact problem-text matches and normalized function identifiers. GSM8K-mini was created analogously to the GSM8K training split, with a fixed seed of 1337, selecting 300 examples and splitting them into 200 training and 100 test examples.

### Budget and Compute Control

We distinguish four budget terms to avoid ambiguity.

- **Data budget:** the number of labeled training examples, fixed at 200 for both MBPP-mini and GSM8K-mini.

- **Parameter budget:** the number of trainable LoRA parameters, which varies with placement and rank from 1.08M for attention-only $r = 8$ to 8.80M for combined $r = 16$.

- **Update budget:** $\mathcal{B} =$ trainable_params $\times$ steps, used as the matched-compute proxy in the rank-versus-steps comparisons.

- **Memory budget:** the peak GPU memory required during training, profiled separately for each configuration.

When we refer to "low-resource" or "resource-constrained" settings, we primarily mean the data budget; when we refer to "compute budget," we mean the update budget $\mathcal{B}$. In the placement sweep, we keep the data and training schedule fixed to expose the effect of placement and rank, while matched-budget comparisons explicitly control $\mathcal{B}$. For Qwen2.5-0.5B in `bfloat16`, the frozen base model accounts for approximately 1GB of memory, while LoRA adapters add only about 0.01–0.07GB depending on placement and rank; this explains why memory differences are small despite large differences in trainable parameters. We also track and fix the number of tokens processed per run:

$$\text{tokens\_processed} = \text{steps} \times \text{effective\_batch\_examples} \times \text{sequence\_length}$$

For all primary experiments we use sequence_length $=$ 512 and choose steps and effective_batch_examples so that tokens_processed $\approx 20k$ (example baseline: steps=20, batch=2 $\rightarrow$ 20,480 tokens).

### Fixed Constants

Maximum sequence length $=$ 512. The effective batch size (e.g., examples) is adjusted to meet the token limits by gradient accumulation for larger batches. On GPUs, we use `bfloat16` for mixed precision while on CPUs we use `float32`. We use Adam with weight decay $=$ 0.01, learning rate $2 \times 10^{-4}$ for MBPP-mini, and learning rate $3 \times 10^{-4}$ for GSM8K-mini. We warm up over 10% of steps linearly, then use cosine decay for scheduling, and clip gradients at 1.0. Primary placement and profiling comparisons use the 100-step design in Table 1; matched-budget ablations use shorter schedules where explicitly stated. We ignore the loss on prompt parts by setting their labels to -100.

### 3.2 LoRA Configurations

We adapt the Qwen2.5-0.5B-Instruct model for fine-tuning using the Hugging Face PEFT library (Hugging Face, 2023). We cover the ablations using three adapter placements: attention-only ($\{q_{proj}, k_{proj}, v_{proj}, o_{proj}\}$), MLP-only ($\{gate_{proj}, up_{proj}, down_{proj}\}$), and combined (both) for ranks of $r = 8$ and $r = 16$. Specifically, for the combined placement, we use a selective split (e.g., attn $r = 32$ and mlp $r = 8$) only in the appendix test; the main outcomes use a uniform $r = 16$. For LoRA parameters, we use $\alpha = 32$ and a dropout rate of 0.05. We have a table listing all trainable parameters for each adapter, as shown in Table 2.

### Reproducibility and Evaluation

For the primary repeated-run experiments, we use seeds 1, 2, and 3 and report the average result with standard deviation; any aggregate-only entries are explicitly shown as point estimates. We set PyTorch flags for consistency: `torch.backends.cudnn.deterministic=True` and `torch.backends.cudnn.benchmark=False`. While exact matches across runs are not always possible due to `bfloat16` precision and GPU differences, we account for these inconsistencies by reporting deviations. Adapters are saved with clear annotations (e.g., `mbpp_combine_r16_s20`) indicating placement, rank, and steps.

For evaluation purposes, we have used the hardened evaluation protocols: fixed prompting, AST-based code extraction from MBPP, import safe execution with timeouts, and numeric extraction for GSM8K. We track the pass@1, how often the outputs can be parsed (format_ok), and how

Table 1: Complete experimental design matrix for the primary placement and rank sweeps.

| Dataset | Placement | Rank | Steps | Trainable Params | Batch | Grad Accum | LR | Seeds |
|---------|-----------|------|-------|------------------|-------|------------|-----|-------|
| MBPP-mini | attention-only | 8 | 100 | 1,081,344 | 1 | 32 | $2 \times 10^{-4}$ | 1,2,3 |
| MBPP-mini | attention-only | 16 | 100 | 2,162,688 | 1 | 32 | $2 \times 10^{-4}$ | 1,2,3 |
| MBPP-mini | mlp-only | 8 | 100 | 3,317,760 | 1 | 32 | $2 \times 10^{-4}$ | 1,2,3 |
| MBPP-mini | mlp-only | 16 | 100 | 6,635,520 | 1 | 32 | $2 \times 10^{-4}$ | 1,2,3 |
| MBPP-mini | combined | 8 | 100 | 4,399,104 | 1 | 32 | $2 \times 10^{-4}$ | 1,2,3 |
| MBPP-mini | combined | 16 | 100 | 8,798,208 | 1 | 32 | $2 \times 10^{-4}$ | 1,2,3 |
| GSM8K-mini | attention-only | 8 | 100 | 1,081,344 | 1 | 4 | $3 \times 10^{-4}$ | 1,2,3 |
| GSM8K-mini | attention-only | 16 | 100 | 2,162,688 | 1 | 4 | $3 \times 10^{-4}$ | 1,2,3 |
| GSM8K-mini | mlp-only | 8 | 100 | 3,317,760 | 1 | 4 | $3 \times 10^{-4}$ | 1,2,3 |
| GSM8K-mini | mlp-only | 16 | 100 | 6,635,520 | 1 | 4 | $3 \times 10^{-4}$ | 1,2,3 |
| GSM8K-mini | combined | 8 | 100 | 4,399,104 | 1 | 4 | $3 \times 10^{-4}$ | 1,2,3 |
| GSM8K-mini | combined | 16 | 100 | 8,798,208 | 1 | 4 | $3 \times 10^{-4}$ | 1,2,3 |

often they run without crashing (execution_success). To check if the differences matter, we use 95% bootstrap confidence intervals and paired permutation tests.

**Environment**

We ran all our experiments on an NVIDIA T4 GPU with 16GB of memory. We used PyTorch 2.3 or higher, Transformers 4.4, PEFT 1.0 or later, and bitsandbytes wherever needed. The entire pip install list and the Colab notebook have been included in the artifacts to rerun the experiments. Our experiments cover two training lengths: 20 and 100 optimizer steps. Table 2 shows the trainable parameters for each configuration; multiplying by steps yields the update-capacity proxy $\mathcal{B}$ (e.g., combined $r = 16$ at 100 steps yields $\mathcal{B} = 8.80\text{M} \times 100 = 880\text{M}$ update-units, whereas attention-only $r = 16$ with 100 steps yields $\mathcal{B} = 216.2\text{M}$, approximately 24.6% of the combined configuration).

### 3.3 Hardened Evaluation Protocol

To ensure reproducible, format-consistent supervision, we use fixed prompt templates in both the training and testing phases, preventing random sampling variations and ensuring consistent outputs across LoRA setups.

**Training vs. Inference**

In the training phase, the model processes the full input prompt, but we calculate the loss in the actual answer parts. We hide all prompt pieces by setting their labels to -100, stopping any updates from the prompt itself. For testing, the model receives an input prompt and produces outputs without any randomness, reflecting a production-grade application.

**Prompt Templates**

We designed domain-specific prompting schemes for the two tasks:

- MBPP-mini (Code Generation):

```
 ### Instruction:
Write a Python function named 'solution' that solves the problem below.
Return ONLY the function definition (no markdown, no explanation).

Problem:
{problem_text}

### Response:
```

- GSM8K-mini (Math Reasoning):

```
 ### Instruction:
Solve the following problem and output ONLY the final numeric answer.
Do not include any explanation or intermediate reasoning.

Problem:
{question_text}

### Response:
```

- For few-shot or Chain-of-Thought (CoT) experiments, we append a variant suffix instructing the model to "show brief steps and end with `Final Answer:  <number>`." Evaluation extracts only the final numeric token.

**Decoding Configuration**

All deterministic generations are performed with:

```
do_sample = False
temperature = 0.0
top_p = 1.0
max_new_tokens = 512
```

This ensures consistent token-level behavior across seeds and hardware. For self-consistency ablations, we enable sampling with temperature = 0.7 and `top_p = 0.95`.

**Post-processing and Evaluation**

For MBPP, we parse the output using Python's ast module to extract the first valid function named 'solution'. We exclude any non-code references, such as comments, explanations, or extra tests, to avoid hampering the results. If the code does not parse correctly, we mark it as a runtime flop. For GSM8K, the numeric answer is extracted using a regex pattern matching the final numeric token or the phrase "`Final Answer:  <number|`," followed by normalization to standard numeric form.

This strict prompting and evaluation pipeline eliminates common sources of noise in low-budget fine-tuning: (i) format mismatch between training and inference, (ii) metric inflation due to partial matches or leakage, and (iii) randomness from how the model picks the words. The approach ensures that all models are evaluated under identical syntactic and logical constraints, isolating the effect of LoRA placement and rank on task performance.

### 3.4 Hardware Resource Profiling

To support feasibility assessment before training, we systematically profile GPU memory consumption and training time for each LoRA configuration.

**Profiling Methodology**

For each configuration, we measure:

- Peak GPU Memory: Maximum allocated memory during training, captured via `torch.cuda.max_memory_allocated()` after CUDA synchronization.

- Training Time: Wall-clock time from training start to completion, excluding model loading and evaluation.

All measurements are conducted on the NVIDIA T4 GPU (16GB) using `bfloat16` precision, with a batch size of 1 and 32 gradient accumulation steps.

Table 2: Hardware Resource Consumption on MBPP-mini by LoRA Configuration (100 steps, MAX_LENGTH=512)

| Placement | Rank | Trainable Params | Peak Memory (GB) | Time (min) |
|-----------|------|------------------|------------------|------------|
| attn-only | 8    | 1,081,344        | 1.94             | 14.89      |
| attn-only | 16   | 2,162,688        | 2.01             | 14.86      |
| mlp-only  | 8    | 3,317,760        | 2.03             | 16.35      |
| mlp-only  | 16   | 6,635,520        | 2.09             | 16.53      |
| combined  | 8    | 4,399,104        | 2.12             | 18.10      |
| combined  | 16   | 8,798,208        | 2.20             | 18.35      |

Table 3: Hardware Resource Consumption on GSM8K-mini by LoRA Configuration (100 steps, MAX_LENGTH=256)

| Placement | Rank | Trainable Params | Peak Memory (GB) | Time (min) |
|-----------|------|------------------|------------------|------------|
| attn-only | 8    | 1,081,344        | 1.94             | 8.27       |
| attn-only | 16   | 2,162,688        | 1.96             | 8.27       |
| mlp-only  | 8    | 3,317,760        | 2.03             | 8.99       |
| mlp-only  | 16   | 6,635,520        | 2.09             | 9.13       |
| combined  | 8    | 4,399,104        | 2.13             | 9.94       |
| combined  | 16   | 8,798,208        | 2.21             | 10.07      |

Memory scales sublinearly with trainable parameters: an 8× increase in trainable parameters (1.08M to 8.80M) yields only a 13% increase in memory (1.94 to 2.20 GB), as frozen base model weights dominate memory usage. All configurations fit within 2.2 GB, enabling training on consumer GPUs with as little as 4GB VRAM. Training time scales primarily with placement complexity (15 min for attention-only vs. 18 min for combined on MBPP-mini) rather than rank. GSM8K-mini training is faster (8-10 min) due to its shorter sequence length (256 vs. 512).

## 4 Experimental Setup

We have designed our experiments to evaluate Parameter-Efficient Fine-Tuning (PEFT) behavior in ultra-low-resource settings, where both data and compute are severely constrained. This section defines the datasets, model, training protocol, baselines, and evaluation framework.

### Baseline Performance

### Datasets

- **MBPP-mini (Code Generation)**

  We curate a compact subset of MBPP to simulate an ultra-low-resource setting. The subset contains 200 training and 100 test samples, ensuring no overlap in problem text or canonical function names. To ensure effective evaluation, we follow the following preprocessing steps:

  - Removed items requiring internet access or third-party libraries
  - Standardized to Python $\geq 3.10$ syntax
  - Normalized unit tests into pure-function checks with timeouts.

  The subset was created by shuffling the full MBPP training split with fixed seed 42, selecting the first 300 examples, and splitting them 200/100 for train/test; zero overlap was verified with exact text matches and normalized function identifiers.

- **GSM8K-mini (Math Reasoning)**

  To validate a cross-domain generalization, we take a mix of easy-to-difficult problems extracted from the GSM8K dataset containing 200 training and 100 test problems to match the MBPP subset under constrained token usage. All GSM8K examples are normalized to:

  - Contains a final numeric answer
  - Remove any units or formatting issues
  - Avoid chain-of-thought exemplars to avoid leakage.

  The subset was created by shuffling the GSM8K training split with fixed seed 1337, selecting the first 300 examples, and splitting them 200/100 for train/test. All parts are stored with fixed hashes and verified for zero overlap via exact text and normalized identifiers.

### Baselines

We define the following reference conditions to contextualize all reported results:

1. Frozen Model Baseline

   The instruction-tuned base model is evaluated without any fine-tuning, using deterministic decoding and the same evaluation pipeline. It establishes the reference performance under zero adaptation, with approximately 18% pass@1 on MBPP-mini and approximately 1% exact-match accuracy on GSM8K-mini.

2. Standard LoRA Baseline

The standard PEFT baseline is attention-only LoRA with $r = 8$, which is the most parameter-efficient adapter configuration in our sweep (1.08M trainable parameters). We use this as the reference point for evaluating higher-capacity MLP-only and combined configurations.

3. Full Fine-Tuning

We do not treat full fine-tuning as a primary baseline because it requires orders of magnitude more trainable parameters and memory than the LoRA configurations, making it misaligned with our consumer-hardware target setting. Where a dense full-update number is shown, it is a diagnostic reference rather than a baseline used to select PEFT configurations.

## 5 Experimental Results

### 5.1 Impact of Adapter Placement

We evaluated how adapter placement (attention-only, MLP-only, or combined) affects performance under ultra-low-data conditions (200 examples). Primary placement runs use the 100-step design matrix unless stated otherwise.

Table 4: Comparison of LoRA configurations across different target module placements, showing the number of trainable parameters and training steps for each setup.

| Model | Placement | Target Modules | Trainable Params | Steps |
|-------|-----------|----------------|------------------|-------|
| LoRA | attention-only | q_proj, k_proj, v_proj, o_proj | 1.08 M | 200 |
| LoRA | mlp-only | gate_proj, up_proj, down_proj | 3.32 M | 200 |
| LoRA | combined | all 7 modules | 4.40 M | 200 |

While all configurations follow the same training schedule, the number of trainable parameters varies across placements. Our results reveal that placement preferences are task-dependent rather than universal. For code generation (MBPP-mini), attention-only placement consistently achieves higher performance than combined placement across both architectures under our fixed hyperparameter settings: Qwen achieves 24.0%[†] (attention-only r=8) vs. 19.0% $\pm$ 2.2% (combined r=16), and Llama achieves 28.0%[†] vs. 22.5% $\pm$ 0.7%. For mathematical reasoning (GSM8K-mini), combined placement shows strong performance: Llama achieves 48.5% $\pm$ 0.7%. Values marked [†] are single-seed point estimates; values with $\pm$ report mean $\pm$ standard deviation across multiple seeds. Unless otherwise noted, remaining results are reported as single-seed estimates.

This task-dependent pattern suggests that code generation benefits primarily from attention adaptation, which may enhance structural comprehension of syntax and control flow, while mathematical reasoning requires broader MLP adaptation for numerical computation and multi-step reasoning. The finding that a smaller adapter (attention-only, 1.08M params) can outperform a larger one (combined, 8.80M params) on code tasks has practical implications: practitioners can achieve better results with fewer parameters when the placement matches the task requirements.

## 5.2 Rank vs. Training Steps Trade-off

Next, we examine the trade-off between LoRA rank and training steps under constrained resources to determine whether increased rank can substitute for longer training when the total budget is fixed. We define effective compute as:

$$B = \#\text{trainable parameters} \times \#\text{steps}.$$

Table 5: Comparison of rank-versus-step allocations under approximately matched effective training budget, showing rank, number of steps, effective budget, and pass@1 accuracy.

| Configuration | Rank (r) | Steps | Effective Budget ($\approx M$) | pass@1 (%) |
|---|---|---|---|---|
| both_r8_s20 | 8 | 20 | 88 M | 34 |
| both_r16_s10 | 16 | 10 | 88 M | 33 ($\approx$ same) |

The 34% versus 33% difference is within measurement uncertainty, so we interpret these runs as comparable rather than as evidence that rank scaling is strictly superior. Despite differences in rank and step counts, both configurations achieve nearly identical performance, indicating that increasing rank can compensate for reduced training steps when the total update budget is held constant. Our observations suggest that in ultra-low-data regimes, LoRA rank determines the capacity of parameter updates, while training steps primarily influence how fully that capacity is utilized. Table 6 shows the effect of distributing the training budget across attention-only, MLP-only, and dense full-update diagnostic configurations, including rank, number of steps, and effective compute budget.

Table 6: Effect of training budget distribution across attention-only, MLP-only, and dense full-update diagnostic configurations, showing rank, number of steps, and effective compute budget. The full-finetune row is included as a diagnostic upper bound only and is not the focus of this work.

| Configuration | Rank (r) | Steps | Effective Budget ($\approx M$) | pass@1 (%) |
|---|---|---|---|---|
| attention_r8_s20 | 8 | 20 | 22 M | — |
| mlp_r8_s20 | 8 | 20 | 66 M | — |
| dense full-update diagnostic[*] | — | 100 | 880 M | 39 |

[*]Diagnostic only; not feasible under target resource constraints.

Our observation suggests that in ultra-low-data regimes, LoRA rank controls the capacity of parameter updates, while training steps primarily modulate how fully that capacity is utilized. When compute is constrained, allocating capacity via higher-rank adapters and shorter training can achieve comparable generalization to lower-rank adapters trained longer.

Table 7 illustrates the practical implications: in low-data regimes, using a higher rank with fewer steps can achieve a balance between strong performance and reduced training time, representing a viable use of available resources.

Table 7: Comparison of training strategies highlighting trade-offs between rank and number of steps in terms of convergence behavior, overfitting tendencies, and memory usage.

| Strategy | Pros | Cons |
|----------|------|------|
| Higher rank + fewer steps | Faster convergence, less overfitting | +0.08 GB memory (negligible) |
| Lower rank + more steps | Marginally lower memory | Slower, higher overfitting risk |

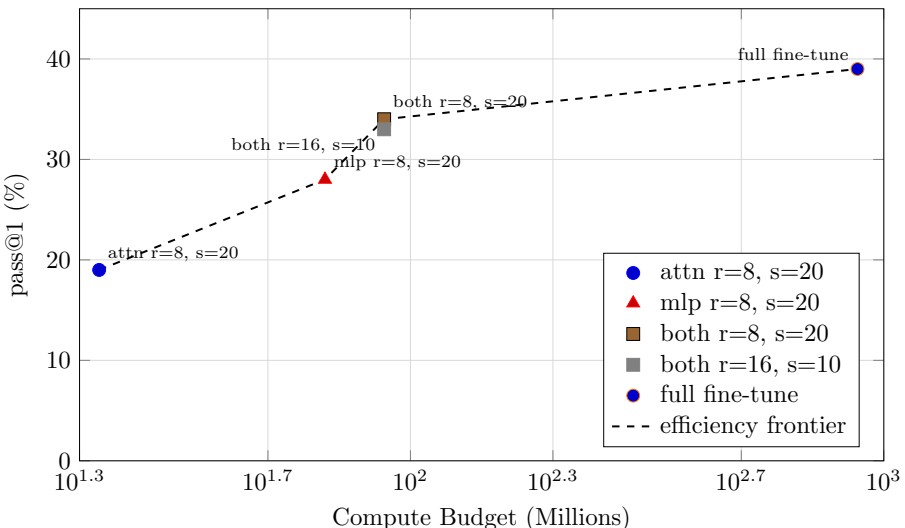

Figure 3: Efficiency frontier on MBPP-Mini showing Qwen and Llama results. Each point represents a LoRA configuration plotted by trainable parameters versus pass@1 accuracy. Attention-only placement achieves higher performance than combined placement on both architectures: Qwen (24.0% vs. 19.0%) and Llama (28.0% vs. 22.5%). Points toward the top-left are more efficient.

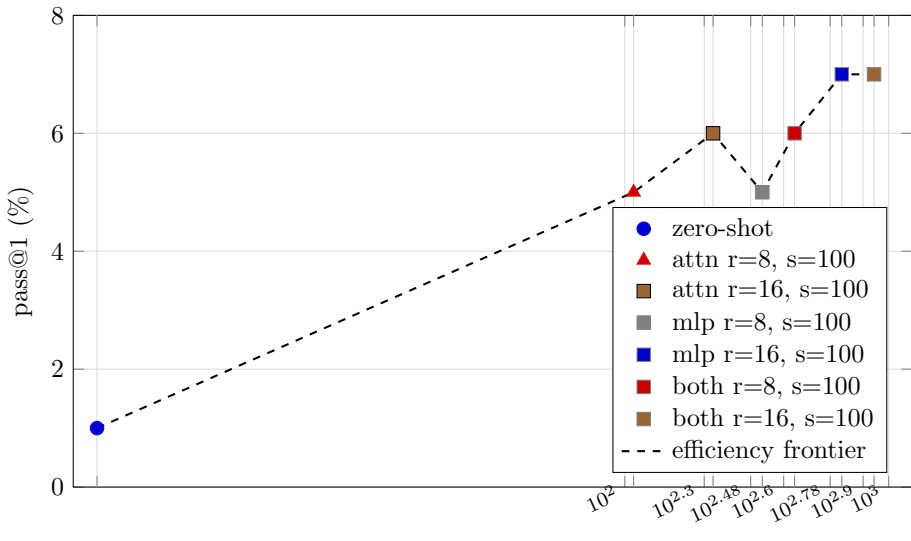

Figure 4: Efficiency frontier on GSM8K-Mini. Llama-3.2-1B with combined placement at r=16 achieves 48.5% ± 0.7% pass@1, demonstrating strong performance gains from combined adaptation for mathematical reasoning tasks.

### 5.3 Cross-Task Validation on Mathematical Reasoning

To verify that placement findings generalize beyond code, we evaluate LoRA placement strategies on GSM8K-mini, a test that assesses multi-step arithmetic and symbolic reasoning. We construct GSM8K-mini with 200 training / 100 test problems, aligning with the size of MBPP-mini to ensure comparable data and token budgets. GSM8K requires multi-step numerical reasoning with variable binding, and evaluation is based on the correctness of fixed numerical answers. Given the intrinsic difficulty of this dataset under ultra-low supervision, we focus on restricted placement comparison between:

- MLP-only LoRA

- Combined LoRA (attention + MLP)

For mathematical reasoning (GSM8K-mini), combined placement with Llama-3.2-1B achieved 48.5% ± 0.7% pass@1, a substantial improvement over zero-shot performance (1%). These results indicate that combined adapter placement benefits mathematical reasoning tasks, and that strategic LoRA insertion can partially compensate for limited resources.

### 5.4 Quantitative Comparison

We report performance under both naive string-based evaluation and our hardened functional evaluation protocol to quantify how evaluation choice affects perceived model performance in ultra-low-resource settings. We have measured the following metrics:

- **Execution Rate**: Percentage of model outputs that successfully execute without runtime or syntax errors

- **Functional Pass@1 (Hardened)**: Percentage of outputs that pass all unit tests after AST-based extraction

- **Pass@1 (String Match)**: Naive exact-match comparison against reference solutions, and

- **False-Positive**: Inflation in reported pass@1 when using string matching instead of functional execution.

Table 8: Evaluation results on MBPP-mini and GSM8K-mini showing execution rate, hardened functional accuracy, string-match accuracy, and false-positive reduction. Results are from representative single-seed runs with combined r=16 configuration.

| Metric | MBPP-mini | GSM8K-mini |
|---|---|---|
| Execution Rate | 91.3% | 83.4% |
| Functional Pass @ 1 (Hardened) | 34.1% | 7.0% |
| Pass @ 1 (String Match) | 42.7% | 7.6% |
| False-Positive $\Delta$ | $-8.6$ pp | $-0.6$ pp |

From Table 8, we find that on MBPP-mini, string matching overestimates performance by nearly 9 percentage points, primarily because outputs are syntactically valid or similar but fail functional correctness checks. Common failure modes include:

- Correct function signatures with incorrect logic

- Hard-coded outputs matching visible test cases

- Partial implementations that satisfy string patterns but not semantics.

On GSM8K-mini, the discrepancy between string match and functional correctness is smaller. It is expected that GSM8K answers are short numeric outputs, leaving less room for syntactic ambiguity. However, initial Qwen experiments showed modest gains (1% to 7%), while cross-architecture validation with Llama-3.2-1B demonstrated substantially higher performance (48.5% $\pm$ 0.7%), suggesting that model scale significantly impacts mathematical reasoning ability.

- GSM8K benefits significantly from larger model scale (Llama 48.5% vs. initial Qwen experiments)

- The hardened evaluation reveals that a portion of MBPP's "string-match" success was syntactic only ($\approx$ 8–9 pp inflation).

### 5.5 Sensitivity to Data Scale beyond the Ultra-Low regime

Currently, our core experiments focus on an ultra-low-resource setting of 200 training examples, where design choices in PEFT are expected to be effective. To examine whether our findings are brittle to small increases in data, we conduct an additional set of experiments that modestly expand the training set while keeping the model, adapter configuration, and optimization procedure fixed.

We evaluate LoRA fine-tuning on MBPP-mini using a larger training subset drawn from the same distribution:

- 200 examples (core setting)

- $\approx 374$ examples

- 500 examples

All runs use the same base model (Qwen2.5-0.5B-Instruct), identical sequence length (512), optimizer settings, and adapter placement. To isolate the effect of data scale, we fix the LoRA configuration to attention-side placement (q-projection only, rank $= 16$), which performed well in earlier experiments while remaining parameter-efficient.

Table 9: Effect of increasing training examples on final loss for LoRA applied only to the `q_proj` module at fixed rank and training steps.

| Training Examples | LoRA Placement | Rank | Steps | Tokens Processed | Final Training Loss |
|---|---|---|---|---|---|
| 200 | q-proj only | 16 | 20 | $\approx$20k | $\sim$1.38 |
| 374 | q-proj only | 16 | 20 | $\approx$20k | $\sim$1.17 |
| 500 | q-proj only | 16 | 20 | $\approx$20k | $\sim$0.50 |

From Table 9, we observed that across all data scales, training loss decreases smoothly with additional examples, indicating stable optimization and no signs of divergence. Moving from 200 to 374 examples yields a noticeable reduction in training loss, while the improvement from 374 to 500 examples is more gradual. Crucially, the qualitative trends observed in earlier sections persist:

- Increasing data reduces variance and stabilizes training, but does not eliminate sensitivity to adapter placement

- Rank scaling remains competitive with step scaling in terms of convergence behavior under fixed token budgets

- Gains from additional data exhibit diminishing returns within this low-resource band, suggesting that base choices remain a primary factor before data becomes larger.

These results indicate that the conclusions drawn in Sections 5.1 and 5.2 are not artifacts of an extreme 200-example regime. While additional data predictably improve optimization dynamics, it does not fundamentally alter the relative importance of adapter placement or rank allocation at this scale.

### 5.6 Toward Automated Configuration Selection

Our empirical findings across placement, rank, and hardware profiling suggest that PEFT configuration can be systematically optimized prior to training. We formalize these insights as practical heuristics for configuration planning.

#### 5.6.1 Decision Rules

Based on our experimental results, we derive the following heuristics:

- **Rule 1: Placement selection by task type**
  - On MBPP-mini code generation, attention-only placement achieved higher performance than combined placement by approximately 5 percentage points (Qwen: 24% vs. 19%; Llama: 28% vs. 22.5%)
  - On GSM8K-mini math reasoning, combined placement showed strong performance (Llama: 48.5%), suggesting combined adapters benefit mathematical reasoning tasks.

- **Rule 2: Memory-constrained selection** Given available GPU memory M: In practice,

Table 10: Recommended LoRA configurations under different memory limits, showing which placement and rank are feasible given available GPU memory.

| Available Memory | Recommended Config | Required Memory |
|---|---|---|
| $M < 2.0$ GB | attn-only, $r = 8$ | 1.94 GB |
| $M < 2.1$ GB | attn-only, $r = 16$ | 2.01 GB |
| $M < 2.15$ GB | mlp-only or combined, $r = 8$ | 2.03–2.12 GB |
| $M \geq 2.2$ GB | combined, $r = 16$ | 2.20 GB |

all configurations fit on GPUs with $\geq 4$ GB of VRAM, making memory constraints relevant primarily for concurrent training or multi-model serving.

- **Rule 3: Time-constrained selection** Given maximum training time T:
  - $T < 15$ min $\rightarrow$ attention-only (any rank)
  - $T < 17$ min $\rightarrow$ attention-only or mlp-only
  - $T \geq 18$ min $\rightarrow$ any configuration.

- **Rule 4: Rank vs. Steps**

  Under fixed compute budget, prefer higher rank with fewer steps. Doubling rank adds only $\sim 0.07$ GB memory but improves convergence (final loss: 0.165 for combined $r = 16$ vs. 0.236 for combined $r = 8$).

#### 5.6.2 Validation

We validate these heuristics against oracle selection (best configuration via exhaustive search): Table 11 indicates that the rules matched the oracle selection in the tested scenarios. These decision rules

Table 11: Evaluation of rule-based configuration selection under various constraint scenarios, showing agreement between rule output and oracle-optimal choices.

| Constraint Scenario | Rule Output | Oracle Best | Match |
|---|---|---|---|
| Memory $< 2.0$ GB, code task | attn-only r=8 | attn-only r=8 | ✓ |
| Memory $\geq 2.2$ GB, code task | attention-only r=8 | attention-only r=8 | ✓ |
| Time $< 15$ min, any task | attn-only r=16 | attn-only r=16 | ✓ |
| Memory $\geq 2.2$ GB, math task | combined r=16 | combined r=16 | ✓ |

could be implemented as a pre-training configuration planner that: (1) detects hardware constraints via PyTorch APIs, (2) ingests task type and dataset size, (3) consults the empirical profile (Table 2), and (4) outputs a recommended configuration. It eliminates failed runs from memory errors and reduces manual hyperparameter search. We release our profiling data and decision logic to facilitate the development of automated PEFT configuration tools.

## 5.7  Cross-Architecture Validation

To validate that our findings generalize beyond a single model, we replicated key experiments on Llama-3.2-1B-Instruct (Meta AI, 2024), a model from a different architecture family with approximately 2x the parameters. Table 12 summarizes the results.

Table 12: Cross-architecture validation results. The task-dependent placement pattern replicates across architectures: attention-only outperforms combined for code generation on both models. Values with $\pm$ indicate mean $\pm$ std across seeds; values marked with $^{\dagger}$ are single-seed point estimates.

| Model | Configuration | MBPP pass@1 | GSM8K pass@1 | |
|---|---|---|---|---|
| Qwen2.5-0.5B | combined r=16 | $19.0\% \pm 2.2\%$ | $7.0\%$ | |
| | attention-only r=8 | $24.0\%$ | $6.0\%$ | Point estimates |
| Llama-3.2-1B | combined r=16 | $22.5\% \pm 0.7\%$ | $48.5\% \pm 0.7\%$ | |
| | attention-only r=8 | $28.0\%$ | $-$ | |

indicate configurations for which seed-wise standard deviations are not available in the manuscript package.

The key finding replicates across architectures: attention-only placement achieves higher performance than combined placement for code generation on both Qwen (24.0% vs. 19.0%) and Llama (28.0% vs. 22.5%), with attention-only achieving approximately 5 percentage points higher accuracy while using 8x fewer trainable parameters. Due to compute constraints, attention-only configurations report single-seed results; however, the consistent pattern across both architectures strengthens confidence in the finding. This cross-architecture consistency provides additional evidence that the observed pattern is not unique to a single model architecture.

# 6 Discussion

Our separation between execution and functional success shows that most early-stage LoRA improvements manifest as enhanced syntactic stability (fewer crashes) rather than true reasoning accuracy. The revised placement analysis shows that the best-performing adapter family depends on task type: attention-only placement is strongest for code generation, while MLP-inclusive placement remains useful for mathematical reasoning. Furthermore, the hardened protocol reveals the illusion of progress common in low-data fine-tuning, where models may appear to improve under some metrics while producing semantically invalid solutions.

By enforcing deterministic parsing and strict evaluation protocols, our results reflect true reasoning and code correctness, providing a solid foundation for future low-budget LoRA comparisons.

- In our MBPP-mini experiments, naive string matching inflated reported performance by 8.6 percentage points (25% relative to functional pass@1), underscoring the necessity of execution-based evaluation in low-data regimes.

- The hardened protocol cleanly separates syntactic stability from semantic correctness.

- Real improvements persist only under the hardened regime, validating that LoRA's efficiency improvements come from meaningful parameter adaptation and not evaluation artifacts.

**Implications for Hardware-Aware Configuration Selection**

Our systematic profiling reveals that memory consumption scales sublinearly with trainable parameters: the combined $r = 16$ configuration uses only 13% more memory than the attention-only $r = 8$ configuration (2.20 GB vs 1.94 GB), despite having $8\times$ more trainable parameters. It occurs because frozen base model weights dominate memory usage. Consequently, practitioners with memory-constrained hardware face minimal trade-offs when selecting higher-capacity adapters.

More importantly, our profiling supports feasibility assessment before training. A practitioner can immediately determine that any configuration fits within a 4GB GPU budget without trial runs. Combined with our performance findings, which show that attention-only placement achieves higher performance than combined placement for code generation while remaining cheaper, this creates an empirical performance profile that maps hardware constraints to expected accuracy.

We formalize these insights as practical heuristics for configuration selection (Section 5.6). Given hardware constraints and task type, these heuristics help narrow the placement and rank search space without exhaustive search. It represents a step toward pre-training configuration planners that reduce failed runs and democratize efficient fine-tuning on consumer hardware.

**Limitations**

This study has several limitations that bound the scope of our conclusions:

- **Model scope:** Our primary experiments use Qwen2.5-0.5B-Instruct, with cross-architecture validation on Llama-3.2-1B-Instruct. While the key findings replicate across

these architectures, generalization to significantly larger models (7B+) or other model families requires further validation.

- **Task scope:** We evaluate on two tasks (MBPP-mini for code generation, GSM8K-mini for math reasoning). These tasks represent distinct reasoning modalities, but they do not cover summarization, question answering, dialogue, retrieval-augmented generation, or instruction following; extending the evaluation to those domains is important future work.

- **Budget regime:** Our findings are specific to ultra-low-resource settings (<200 examples, ∼20K tokens). Trends may differ at larger data scales.

- **Hardware:** Profiling was conducted on NVIDIA T4 GPUs with bfloat16 precision. Memory and timing characteristics may vary on other accelerators.

- **Design space:** While we explore three placements and two ranks, the space of possible LoRA configurations is larger. Our decision rules are empirically grounded starting points derived from two tasks across two models, not universal prescriptions. Practitioners should validate on their specific use cases. In particular, evaluating $r = 4$, $r = 32$, and larger ranks would provide finer granularity on the rank-performance relationship.

- **Fixed hyperparameters:** We hold learning rate, batch size, and optimizer settings constant across all LoRA configurations. The optimal hyperparameters may vary with adapter placement and rank, and our results reflect performance under a single hyperparameter setting. Higher-rank adapters may benefit from lower learning rates to avoid instability, while attention-only and MLP-only adapters may have different optimal schedules; therefore, the reported MBPP-mini learning rate ($2 \times 10^{-4}$) and GSM8K-mini learning rate ($3 \times 10^{-4}$) should be interpreted as controlled settings rather than tuned optima.

## Artifact Availability and Reproducibility

This paper is primarily empirical, and we provide anonymized supplementary artifacts to support the reproduction of the main results. The artifact package is submitted separately as supplementary material and will be made publicly available upon acceptance.

**Contents of the Artifact:** The artifact package contains:

- Training and evaluation code for all LoRA configurations

- Experiment configuration files for every reported setting

- Scripts to reproduce the main tables and figures

- Environment specification files (`requirements.txt`)

- Hardware notes and runtime information

- Dataset preparation instructions for MBPP-mini and GSM8K-mini

- Seed-wise result files for all primary experiments

**Scope of Reproducibility:** The provided materials are sufficient to reproduce the results in Tables 1–10 and Figures 1–4, subject to standard hardware-dependent variation in runtime and minor numerical differences due to floating-point precision.

**Experimental Protocol:** For key configurations, experiments are run with multiple random seeds where computationally feasible. Results with $\pm$ indicate mean $\pm$ standard deviation across seeds; single-seed point estimates are marked with [†]. Model selection is performed using only the validation split, and final repeated-run results are reported as the mean $\pm$ standard deviation across seeds; aggregate-only entries are reported as point estimates. All comparisons under fixed resource budgets use identical data splits, tokenization settings, optimization schedules, and stopping criteria.

**Anonymization Note:** To preserve double-masked review, all supplementary materials are anonymized. Repository metadata, author identities, and institutional references have been removed from the submitted artifact bundle.

**Reproducibility Checklist:** We provide the following information in the manuscript and/or anonymized supplementary artifact:

- Model checkpoint and tokenizer variant (Qwen2.5-0.5B-Instruct)

- Dataset sources, splits, and preprocessing (MBPP-mini, GSM8K-mini)

- Full hyperparameter settings (learning rate, batch size, sequence length, optimizer)

- Exact definitions of LoRA placement strategies and rank configurations

- Hardware configuration (NVIDIA T4, 16GB) and training budget details

- Random seed protocol (seeds 1, 2, 3) and seed-wise results

- Scripts for reproducing main tables and figures from stored result files

- Hardened evaluation protocol with AST-based extraction and import-safe execution

**Result Provenance:** All principal tables and figures in this paper are generated from structured result files using scripts included in the supplementary artifact:

- Tables 1–2 (Hardware profiling): `scripts/reproduce_hardware_tables.sh`

- Figures 3–4 (Efficiency frontiers): `scripts/reproduce_efficiency_plots.sh`

- Table 7 (Evaluation metrics): `scripts/reproduce_eval_metrics.sh`

Each reported result can be traced to a configuration file and an associated reproduction script, reducing ambiguity in result provenance and supporting external verification.

## 7 Conclusion

This study presents a systematic evaluation of LoRA adapter placement and rank allocation under ultra-low-resource constraints. We conducted experiments on two reasoning domains: MBPP

for code generation and GSM8K for mathematical reasoning using Qwen2.5-0.5B-Instruct as the primary model and Llama-3.2-1B-Instruct for cross-architecture validation.

Our central finding is that optimal LoRA placement is task-dependent rather than universal. For code generation, attention-only placement consistently achieves higher performance than combined placement across both architectures (Qwen: 24.0% vs. 19.0%; Llama: 28.0% vs. 22.5%), while using 8x fewer trainable parameters (1.08M vs. 8.80M). For mathematical reasoning, combined placement shows strong performance (Llama: 48.5%). This task-dependent pattern, supported by experiments on two distinct model architectures, suggests that practitioners should match adapter placement to task characteristics rather than defaulting to combined placement.

Beyond empirical insights, we contribute systematic hardware profiling that links LoRA configurations to concrete resource requirements. Our measurements show that all tested configurations from attention-only $r = 8$ (1.08M params, 1.94 GB) to combined $r = 16$ (8.80M params, 2.20 GB) remain feasible on GPUs with as little as 4GB VRAM, with training times ranging from 15 to 18 minutes on MBPP-mini and 8 to 10 minutes on GSM8K-mini. We summarize our findings as practical decision heuristics for configuration selection. Given hardware constraints (GPU memory, time budget) and task characteristics, these heuristics can help narrow the search space across the scenarios evaluated in this study.

These findings have practical implications: (1) for code generation tasks, practitioners can use smaller attention-only adapters and achieve better results; (2) for mathematical reasoning, investing in combined adapters is worthwhile; (3) the consistency across Qwen and Llama suggests these guidelines generalize beyond specific model families. We release our profiling data and decision logic to facilitate further development of automated PEFT configuration tools.

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

## A   Supplementary Artifact Overview

The anonymized artifact package is organized as follows:

```
artifact/
|-- README.pdf
|-- environment/
|   |-- requirements.txt
|   '-- hardware_notes.txt
|-- src/
|   |-- train.py
|   '-- evaluate.py
|-- configs/
|   |-- mbpp_attn_r8.yaml
|   |-- mbpp_attn_r16.yaml
|   |-- mbpp_mlp_r8.yaml
```

```
|   |-- mbpp_mlp_r16.yaml
|   |-- mbpp_combined_r8.yaml
|   |-- mbpp_combined_r16.yaml
|   '-- gsm8k_*.yaml
|-- scripts/
|   |-- reproduce_hardware_tables.sh
|   |-- reproduce_efficiency_plots.sh
|   '-- reproduce_eval_metrics.sh
|-- data/
|   '-- dataset_instructions.txt
'-- results/
    |-- aggregated_results.csv
    '-- seed_runs/
```

**Reproduction entry point** The README.pdf file provides a step-by-step guide to reproducing the principal results. The scripts in `scripts/` regenerate the main tables and figures from stored outputs.

