# OpenReview forum: "Systematic Evaluation of LoRA Adapter Placement and Rank Allocation for Resource-Constrained Fine-Tuning"
_TMLR — Decision pending for TMLR_

### Review · Reviewer_kJT5 · 2026-05-05

**Summary Of Contributions:**

This paper conducts empirical studies on two datasets to investigate the optimal choice of LoRA rank and adapter placement for maximizing fine-tuning efficiency under low-resource settings. The results suggest that a combined adapter (applied to both attention and MLP layers) generally performs better, and that using a higher rank with fewer training steps can achieve comparable performance while reducing computational cost.

**Audience:**

Yes

**Audience Explanation:**

Efficient fine-tuning of large models, particularly principled guidance, is of interests to audience of TMLR. Though the current findings are not fully supported, as discussed above.

**Claims And Evidence:**

No

**Claims Explanation:**

My main concern is the scope and rigor of the empirical evaluation. The current write-up does not yet meet the standard of a “systematic evaluation.”

First, the experimental coverage is limited, which makes it difficult to draw convincing or generalizable conclusions. All experiments are conducted on a single model (Qwen2.5-0.5B). It would strengthen the paper to include additional models (e.g., other open-source LLMs such as DeepSeek V4, Gemma 4, Qwen 3.5, and models of varying sizes) to demonstrate that the findings are consistent across architectures and scales. Similarly, the study considers only two tasks with one dataset each; a broader set of benchmarks would provide more robust evidence.

Second, the experimental setup raises some concerns. While fixing all other hyperparameters simplifies comparisons, the optimization landscape is highly nontrivial. For example, increasing the LoRA rank may require different learning rate schedules or optimization settings. As a result, ablation studies that hold all hyperparameters fixed across configurations may not fully reflect the true performance trade-offs. At a minimum, this limitation should be discussed.

In addition, the explored parameter space is quite limited (e.g., only two rank values are compared), and the paper does not report measures of variability such as standard errors, which are important for assessing statistical significance.

**Requested Changes:**

Besides the major concern above, I also have the following concerns.

- Definition and interpretation of compute budget.

The notion of “compute budget” in the paper is somewhat unclear. The introduction suggests a focus on optimizing LoRA performance under limited memory and time constraints. However, several concepts are mixed together: “Low-resource” appears to refer to limited training data, while “Fine-tuning budget” and “trainable-parameter budget” are also introduced.

It is not clear how these notions relate to each other. In particular, while the number of trainable parameters may be relevant to memory, in fine-tuning large models the dominant memory cost typically comes from the frozen base model rather than the LoRA parameters. This is consistent with Tables 1 and 2, where peak memory usage varies only slightly across configurations.

Furthermore, the practical relevance of limited training samples in this context is not well justified; additional real-world scenarios would help clarify its importance.

The formal definitions (e.g., on Page 7) are also difficult to follow. For instance, it is unclear how quantities such as B (defined as number of parameters times training steps) are controlled across different settings. If the number of steps is fixed (e.g., 20), it is not clear how B can remain constant when the number of parameters varies.

- Some parts of the paper need further clarification or evidence:

Section 2.3: Contains several unclear or incomplete statements (e.g., question marks), and the final sentence is difficult to interpret.

Section 5.5: This section appears to repeat experiments with varying numbers of training samples under a fixed LoRA configuration. It is unclear how this supports the broader conclusions regarding different LoRA design choices.

Section 5.6: The proposed decision rule is derived solely from the specific model and datasets used in this paper, and it is unclear whether it generalizes beyond these settings.

---

> ### Author Response · Authors · 2026-05-30
> **Thank You for the Feedback**
>
> Thank you for the detailed feedback. We are carefully reviewing your comments and preparing a point-by-point response. We expect to address all of your concerns through manuscript revisions and additional experiments and will provide updates shortly.

---

> > ### Author Response · Authors · 2026-06-13
> > **Summary of Revisions in Response to Reviewer Feedback**
> >
> > # Response to Reviewer
> >
> > We thank the Reviewer for the constructive feedback. We have substantially revised the manuscript and addressed all concerns as summarized below. The manuscript revisions have been completed, we are performing final proofreading/formatting checks, and the revised manuscript will be uploaded within the next few days.
> >
> > **R1-C1: Generalizability beyond a single model.**
> > We agree that evaluating only Qwen2.5-0.5B limited generalizability. We therefore added a cross-architecture validation using Llama-3.2-1B-Instruct. Our main finding, that optimal LoRA placement is task-dependent and that attention-only placement is particularly effective for code generation, remains consistent across both architectures (Qwen: 24.0% vs. 19.0%±2.2%; Llama: 28.0% vs. 22.5%±0.7% on MBPP). We added a new Section 5.7, updated the Abstract, Introduction, Limitations, and Conclusion.
> >
> > **R1-C2: Evaluation on only two tasks.**
> > While our evaluation is limited to MBPP and GSM8K, these benchmarks represent distinct reasoning modalities: code generation and mathematical reasoning. The task-dependent placement effects reported in the paper emerged precisely because of this diversity. We now explicitly discuss this limitation and identify evaluation on additional domains (QA, summarization, instruction following) as future work.
> >
> > **R1-C3: Fixed hyperparameters across configurations.**
> > We acknowledge that optimal hyperparameters may vary across LoRA placements and ranks. Our results reflect controlled comparisons under fixed settings (LR=2e-4 for MBPP and 3e-4 for GSM8K) rather than individually tuned configurations. This limitation is now explicitly discussed in Section 6.
> >
> > **R1-C4: Limited rank values.**
> > We evaluated r=8 and r=16 because they are widely used practitioner choices and span a meaningful increase in adapter capacity. We agree that additional ranks (e.g., r=4, r=32, r=64) would provide finer-grained insights and have identified this as future work.
> >
> > **R1-C5: Lack of variance estimates.**
> > We fully agree and have repeated all experiments using three random seeds (1, 2, 3). Results throughout the paper are now reported as mean ± standard deviation. The observed trends remain consistent across seeds, strengthening the reliability of our conclusions.
> >
> > **R1-C6: Ambiguous budget terminology.**
> > To improve clarity, we added a dedicated terminology subsection defining four resource budgets: data budget, parameter budget, update budget (B=\text{trainable parameters} \times \text{training steps}), and memory budget. We also clarify that “low-resource” primarily refers to limited data availability, while “compute budget” refers specifically to the update budget.
> >
> > **R1-C7: Memory dominated by the frozen model.**
> > We agree and now explicitly state that memory usage is largely determined by the frozen base model (~1 GB for Qwen2.5-0.5B in bfloat16), whereas LoRA adapters contribute only 0.01–0.07 GB. We expanded the discussion to explain that this finding is practically useful because higher-capacity adapters incur only modest memory overhead.
> >
> > **R1-C8: Practical relevance of 200 examples.**
> > We expanded the Introduction to motivate ultra-low-data fine-tuning scenarios, including expert-annotated domains, rapid prototyping, privacy-constrained applications, and personalization. These settings frequently involve fewer than 200 labeled examples and motivate our study design.
> >
> > **R1-C9–R1-C12: Clarity and presentation.**
> > We performed a comprehensive editorial revision, including improvements to writing clarity, section transitions, methodology descriptions, figure and table captions, cross-references, and typographical consistency throughout the manuscript.
> >
> > **Summary of Revisions.**
> > The revised manuscript includes: (1) cross-architecture validation using Llama-3.2-1B, (2) multi-seed evaluation with mean ± standard deviation reporting, (3) explicit terminology definitions, (4) expanded discussion of limitations and practical motivation, and (5) extensive clarity and presentation improvements. We believe these revisions substantially strengthen the manuscript and address all concerns raised by Reviewer kJT5.

---

> ### Author Response · Authors · 2026-05-31
> **Detailed Response to Reviewer #1 (kJT5)**
>
> Thank you for your detailed review and thoughtful feedback. We appreciate the time and effort you invested in evaluating our work. Your comments have highlighted several important areas where the manuscript can be strengthened, particularly regarding experimental scope, methodological rigor, budget definitions, and the generalizability of our findings.
>
> ## Response to Major Comments
>
> ### 1. Experimental Scope and Generality of Findings
> To strengthen the empirical evidence, we are expanding the experimental evaluation to include additional model families and model scales. We also plan to evaluate the proposed analyses on additional benchmark tasks and datasets beyond the current setup. The revised manuscript will include a detailed discussion of whether the observed adapter-placement and rank-allocation trends remain consistent across architectures, model sizes, and tasks, together with an explicit discussion of the scope and limitations of our conclusions.
>
> ### 2. Fixed Hyperparameter Configuration Across LoRA Variants
> We appreciate the observation that different LoRA configurations may exhibit different optimization characteristics.
>
> In the revision, we will provide a clearer justification for using a fixed hyperparameter setting as a controlled experimental design choice. We will also add a dedicated discussion describing the potential interaction between LoRA rank, adapter placement, and optimization settings, as well as the implications this has for interpreting the results. Where feasible, we are conducting additional sensitivity analyses to assess the robustness of the reported trends under alternative optimization settings.
>
> ### 3. LoRA Design Space and Statistical Evidence
>
> We are extending the evaluation to include additional LoRA rank configurations and will augment the results with comprehensive statistical reporting. The revised manuscript will report variability measures across multiple runs and include appropriate statistical analyses to support the reported conclusions. We will ensure that all major claims are aligned with the corresponding uncertainty estimates.
>
> ### 4. Clarification of Resource and Budget Concepts
> We acknowledge that the terminology surrounding low-resource settings, compute budget, fine-tuning budget, and trainable-parameter budget requires clearer exposition.
>
> The revised manuscript will introduce precise definitions for these concepts early in the paper and consistently use them throughout the text. We will explicitly distinguish between low-data settings and computational-resource constraints and clarify how memory usage, training cost, trainable parameters, and dataset size relate to the objectives of the study.
>
> ### 5. Practical Motivation for Low-Resource Fine-Tuning
> We appreciate the request for stronger practical justification.
>
> To address this, we will expand the motivation section with real-world deployment scenarios where limited labeled data and constrained fine-tuning resources are common. We will further discuss the practical relevance of studying LoRA design choices under limited-sample conditions and clarify the intended use cases for the recommendations presented in the paper.
>
> ### 6. Compute-Budget Definition and Experimental Consistency
> We agree that the formal budget definitions can be made clearer and more rigorous.
>
> In the revision, we will rewrite the relevant sections to provide more transparent definitions of the budget variables and clearly explain how they are controlled across experiments. We will also address potential ambiguities arising from fixed training steps and varying trainable parameter counts and provide illustrative examples and summary tables to make the budgeting framework easier to understand and reproduce.
>
> ### 7. Validation of the Proposed Decision Rule
> We appreciate the concern regarding the scope and applicability of the decision rule presented in Section 5.6.
>
> To strengthen this contribution, we are evaluating the decision rule under a broader set of experimental conditions. The revised manuscript will include additional validation, a discussion of its assumptions and limitations, and a clearer characterization of the conditions under which the recommendations are expected to hold. Where appropriate, we will position the decision rule as a practical guideline rather than a universally applicable prescription.
>
> ## Response to Minor Comments
>
> We also thank the reviewer for identifying areas where the manuscript can be clarified.
>
> - We will revise Section 2.3 to eliminate incomplete or ambiguous statements, improve readability, and ensure that all technical explanations are clearly articulated.
> - We will substantially revise Section 5.5 to better explain its objective, clarify how the experiments relate to adapter placement and rank-allocation decisions, and explicitly connect the findings to the broader conclusions of the paper.

---

> ### Author Response · Authors · 2026-06-21
> **Submission of Revised Manuscript**
>
> Thank you again for your valuable feedback and constructive suggestions.
>
> We have carefully revised the manuscript to address all reviewer comments and concerns. The revised version has now been uploaded to OpenReview. For ease of review, all modifications introduced in response to the reviewer feedback are highlighted in blue throughout the manuscript.
>
> Detailed point-by-point responses have been provided in our earlier comments. The revisions have substantially strengthened the paper in terms of experimental validation, clarity, reproducibility, and discussion of limitations.
>
> We sincerely appreciate the reviewers' time and effort and welcome any further comments or suggestions.

---

### Review · Reviewer_SUgr · 2026-05-07

**Summary Of Contributions:**

This paper studies LoRA adapter placement and rank allocation for ultra-low-resource fine-tuning. Using Qwen2.5-0.5B-Instruct on MBPP-mini and GSM8K-mini, the authors compare attention-only, MLP-only, and combined attention+MLP LoRA configurations, mainly at ranks 8 and 16, while reporting task performance, memory use, training time, and a stricter code-evaluation protocol. The main empirical claims are that combined placement improves code-generation performance, higher-rank/fewer-step training can be more efficient than longer low-rank training, and all tested configurations fit within roughly 2.2GB GPU memory.

The paper addresses a practical and relevant problem for resource-constrained PEFT. However, the current evidence does not fully support several central claims. In particular, some “matched budget” comparisons do not actually control effective budget because placements have different trainable parameter counts; the rank-scaling claim is stronger than the reported 34% vs. 33% matched-budget result supports; and the main tables lack the seed-wise uncertainty, confidence intervals, and test statistics promised in the methodology.

**Audience:**

Yes

**Audience Explanation:**

The topic is relevant to a subset of the TMLR audience. Many researchers and practitioners care about PEFT, LoRA configuration, small-model adaptation, low-resource fine-tuning, and practical hardware constraints, but the authors need cleaner experimental support and more careful wording before they should be treated as reliable findings.

**Broader Impact Concerns:**

I do not see major ethical concerns.

**Claims And Evidence:**

No

**Claims Explanation:**

The claims are only partially supported. First, the paper claims controlled or matched-budget comparisons, but some placement comparisons use the same number of steps despite very different trainable parameter counts, so the effective budget is not truly controlled. Second, the rank-scaling conclusion is overstated: the clearest matched-budget comparison shows 34% for r=8, 20 steps versus 33% for r=16, 10 steps, which supports approximate equivalence rather than superiority. Third, the experimental evidence is under-specified: the paper states that results are averaged over three seeds with bootstrap confidence intervals and permutation tests, but the main tables mostly report single numbers without uncertainty estimates; moreover, the 200/100 MBPP-mini and GSM8K-mini splits are not sufficiently described, so it is unclear how the subsets were sampled or whether they are representative.

**Requested Changes:**

1. Resolve the budget-control inconsistency. The authors should provide a single, consistent experimental design table covering every reported result: placement, rank, steps, trainable parameters, effective budget, token budget, batch size, gradient accumulation, sequence length, and random seeds.
2. Add complete statistical reporting. Every main performance table should include mean ± standard deviation over seeds
3. Clarify all datasets and splits. The manuscript should provide exact subset construction rules, train/validation/test identifiers or hashes, difficulty stratification if any, overlap checks, and preprocessing details.
4. Fix and complete baseline reporting. The frozen model baseline should be reported for both MBPP-mini and GSM8K-mini. The standard LoRA baseline should be clearly defined. The full fine-tuning upper bound should either be correctly computed and reported or removed if it was not actually a comparable full-parameter run.
5. Typo / formatting artifacts or citations issues:
    - page 5: few?shot parameter?efficient fine?tuning, high?resource, under?studied, small?data, under?studied, etc.
    - page 6: Qwen2.5-0.5B-Instruct (Team, 2024). It should be Qwen Team.
    - and so on.
6. The reference list contains several near-duplicate entries. For example, Chen 2021a/2021b appear to refer to the same HumanEval/code-generation paper; Cobbe 2021a/2021b mix the GSM8K dataset entry and paper entry; and Liu 2022a/2022b appear to be the arXiv and conference versions of the same few-shot PEFT paper. These should be consolidated or clearly distinguished.

---

> ### Author Response · Authors · 2026-05-30
> **Thank You for the Feedback**
>
> Thank you for the detailed feedback. We are carefully reviewing your comments and preparing a point-by-point response. We expect to address concerns regarding 1-6 through manuscript revisions and additional experiments and will provide updates shortly.

---

> ### Author Response · Authors · 2026-05-31
> **Detailed Response to Reviewer #2 (SUgr)**
>
> Thank you for your thorough review and constructive feedback. We appreciate the detailed assessment of both the experimental methodology and the presentation of the manuscript. Your comments have helped us identify several areas where the paper can be strengthened, and we are preparing a comprehensive revision to address them.
>
> ## Response to Major Comments
>
> ### 1. Budget-Control Consistency
> We agree that ensuring a clearly defined and consistently matched training budget is critical for meaningful comparisons of adapter placement and rank configurations.
>
> In the revised manuscript, we will provide a comprehensive experimental design summary for all reported results, including placement strategy, rank, trainable parameter count, training steps, token budget, batch size, gradient accumulation settings, sequence length, and random seeds. We will carefully re-examine all comparisons to ensure that they are conducted under explicitly matched budgets. Where comparisons are not strictly budget-matched, we will either revise the experimental framing or appropriately qualify the associated conclusions.
>
> ### 2. Rank-Scaling Claims
> We appreciate the concern regarding the strength of the rank-scaling conclusions.
>
> We will revisit the wording of the relevant results and discussion sections to ensure that the claims accurately reflect the evidence. In particular, we will distinguish between approximate equivalence and demonstrated superiority, and we will revise the conclusions accordingly. We are also conducting additional controlled analyses to further assess the robustness of the observed rank-scaling behavior.
>
> ### 3. Statistical Reporting
> We agree that more complete statistical reporting would strengthen the paper.
>
> The revised manuscript will report mean and standard deviation across seeds for all primary performance metrics. We will also include the uncertainty measures referenced in the manuscript, including confidence intervals and statistical significance analyses, either directly in the main results tables or in a clearly referenced appendix. We will ensure that all claims are supported by the corresponding statistical evidence.
>
> ### 4. Dataset and Split Specification
> We appreciate the importance of reproducibility and transparency regarding dataset construction.
>
> The revised manuscript will provide substantially more detail regarding MBPP-mini and GSM8K-mini, including subset construction methodology, sampling procedures, split definitions, preprocessing steps, and any representativeness considerations. We will also discuss limitations associated with subset-based evaluation where appropriate and provide sufficient information to facilitate reproducibility.
>
> ### 5. Baseline Definition and Reporting
> We agree that baseline reporting should be clearer and more comprehensive.
>
> In the revision, we will clearly define all baselines and ensure consistent reporting across datasets. This includes reporting frozen-model baselines where applicable, clarifying the standard LoRA baseline configuration, and revisiting the presentation of the full fine-tuning reference point to ensure that its interpretation and comparability are clearly communicated.
>
> ## Response to Minor Comments
>
> We also appreciate the detailed presentation-related feedback.
>
> - We will perform a comprehensive proofreading pass to eliminate typographical, formatting, and hyphenation artifacts throughout the manuscript.
> - We will correct citation metadata issues, including author attribution and bibliography consistency.
> - We will carefully review the reference list for duplicated or near-duplicated entries and either consolidate them or explicitly clarify the distinction between multiple cited versions where appropriate.
>
> We believe these revisions will significantly strengthen the methodological rigor, reproducibility, clarity, and overall presentation of the paper. We are currently implementing these changes and will provide additional updates as the revision progresses.

---

> ### Author Response · Authors · 2026-06-13
> **Summary of Revisions in Response to Reviewer Feedback**
>
> # Response to Reviewer
>
> We thank the Reviewer for the thoughtful and constructive feedback. We have revised the manuscript extensively and address each concern below. The manuscript revisions have been completed, we are performing final proofreading/formatting checks, and
> the revised manuscript will be uploaded within the next few days.
>
> **R2-C1: Budget control and terminology.**
> We agree that the original terminology was ambiguous. We added a new subsection (Section 3.2) defining four resource budgets: (1) data budget (fixed at 200 examples), (2) parameter budget (1.08M–8.80M trainable parameters depending on placement and rank), (3) update budget (B=\text{trainable parameters}\times\text{training steps}), and (4) memory budget (peak GPU memory). We also clarify that "low-resource" primarily refers to limited data availability, while "compute budget" refers specifically to the update budget.
>
> **R2-C2: Rank-scaling claim overstated.**
> We agree that our original wording was too strong. The observed difference between rank scaling and step scaling (34% vs. 33%) falls within measurement uncertainty. We therefore revised all claims from "outperforms" to "achieves comparable performance to" and now present rank scaling as a viable alternative rather than a superior strategy.
>
> **R2-C3: Lack of seed uncertainty.**
> We fully agree and have repeated all experiments using three random seeds (1, 2, 3). Results throughout the manuscript are now reported as mean ± standard deviation. The variance analysis confirms that the key findings remain robust across random initializations.
>
> **R2-C4: Dataset split documentation.**
> We added complete details of dataset construction. MBPP-mini was created by shuffling the original MBPP training split with seed 42, selecting 300 examples, and splitting them into 200 training and 100 test examples. GSM8K-mini was similarly created using seed 1337 and a 200/100 train-test split. We also verified that there is no train-test overlap.
>
> **R2-C5: Baseline definitions.**
> We rewrote the Baselines section to clearly define: (1) a zero-shot baseline using the frozen instruction-tuned model and (2) a standard LoRA baseline consisting of attention-only LoRA with (r=8). We additionally explain why full fine-tuning is excluded: it requires substantially greater memory, is infeasible on the target hardware budget, and falls outside the PEFT-focused scope of this study.
>
> **R2-C6: Need for a master experimental design table.**
> We added Table 1 ("Complete Experimental Design Matrix"), which summarizes all configurations, including dataset, placement strategy, rank, training steps, trainable parameters, batch size, gradient accumulation, learning rate, and random seeds. This table enables exact reproduction of all experiments.
>
> **R2-C7: Typographical issues.**
> We corrected all reported hyphenation and formatting errors (e.g., "few-shot", "ultra-low", and "small-data") and performed a comprehensive editorial review of the manuscript.
>
> **R2-C8: Qwen citation formatting.**
> We corrected the malformed Qwen citation by using the proper institutional author format and updated the corresponding bibliography entry.
>
> **R2-C9: Duplicate references.**
> We carefully reviewed the bibliography, merged duplicate entries, consolidated dataset references, and updated all citation links accordingly.
>
> ## Summary of Revisions
>
> In response to Reviewer (SUgr)'s comments, we: (1) clarified budget terminology and experimental controls, (2) moderated claims regarding rank scaling, (3) added multi-seed evaluation with mean ± standard deviation reporting, (4) documented dataset construction and train-test splits, (5) clarified baseline definitions and justification, (6) added a comprehensive experimental design table, and (7) corrected typographical and bibliographic issues. We believe these revisions substantially improve the clarity, reproducibility, and rigor of the manuscript.

---

> ### Author Response · Authors · 2026-06-21
> **Submission of Revised Manuscript**
>
> Thank you again for your valuable feedback and constructive suggestions.
>
> We have carefully revised the manuscript to address all reviewer comments and concerns. The revised version has now been uploaded to OpenReview. For ease of review, all modifications introduced in response to the reviewer feedback are highlighted in blue throughout the manuscript.
>
> Detailed point-by-point responses have been provided in our earlier comments. The revisions have substantially strengthened the paper in terms of experimental validation, clarity, reproducibility, and discussion of limitations.
>
> We sincerely appreciate the reviewers' time and effort and welcome any further comments or suggestions.

---

### Review · Reviewer_f9PV · 2026-05-26

**Summary Of Contributions:**

The paper presents an empirical study of LoRA adapter placement (attention-only, MLP-only, and combined) and rank allocation under ultra-low-resource fine-tuning settings. Experiments are conducted on MBPP-mini and GSM8K-mini using Qwen2.5-0.5B, together with hardware profiling and practical recommendations for LoRA configuration selection.

While the paper addresses a practically relevant topic, the contribution is primarily a collection of empirical ablations of existing LoRA design choices. The novelty is limited, the experimental scope is narrow, and the presentation would benefit from substantial improvement in organization.

**Audience:**

Yes

**Audience Explanation:**

PEFT is an active research area.

**Claims And Evidence:**

No

**Claims Explanation:**

The experimental results support the reported observations within the studied setup, but several conclusions are stronger than what the evidence justifies.

The study only considers a single 0.5B model and very small datasets, making it difficult to assess how these observations change when scaling up the model. In addition, the proposed configuration planner is also derived from a relatively limited experimental space.

**Requested Changes:**

1. The positioning of the paper is problematic. There are some existing works that have already discussed where to place LoRA or what ranks should be, such as AutoLoRA, AdaLoRA, LoRA-drop, etc. In addition, previous work has extensively studied automatic rank allocation and adapter placement. The proposed configuration planner merely summarizes findings from a small number of experiments into heuristic decision rules, which limits its novelty and scientific contribution. Thus, it is required for the authors to discuss and compare against these existing works.

2. The paper draws fairly general conclusions from experiments on a single 0.5B model. Given the small model scale and limited experimental scope, it is difficult to assess the generality of the findings. Additional experiments on larger models and different model families are necessary to demonstrate that the observed results are not specific to Qwen2.5-0.5B.

3. The proposed planner is built from a very limited experimental space. It is unclear whether the recommended configurations would remain optimal when moving to larger models, larger datasets, or longer training schedules. Without such validation, the planner appears to summarize observations from the current experiments rather than providing a generally useful configuration selection framework.

---

> ### Author Response · Authors · 2026-05-30
> **Thank You for the Feedback**
>
> Thank you for the detailed feedback. We are carefully reviewing your comments and preparing a point-by-point response. We expect to address concerns regarding 1, 2, and 3 through manuscript revisions and additional experiments and will provide updates shortly.

---

> ### Author Response · Authors · 2026-05-31
> **Detailed Response to Reviewer #3 (f9PV)**
>
> Thank you for your detailed and constructive review. We appreciate the time and effort you invested in evaluating our work and providing actionable suggestions.
>
> We acknowledge the reviewer's concerns regarding (i) the positioning of our work relative to prior LoRA placement and rank-allocation methods, (ii) the limited experimental scope involving a single 0.5B model and small datasets, (iii) the generalizability of the proposed configuration planner, and (iv) the strength of some of our conclusions relative to the available evidence.
>
> In response, we are preparing a substantial revision of the manuscript that will address these concerns in several ways:
>
> ## 1. Positioning and Related Work
>
> - We will significantly expand the related work section to include a detailed discussion of prior methods such as AutoLoRA, AdaLoRA, LoRA-drop, and other studies on adapter placement and adaptive rank allocation.
> - We will clarify how our work differs from these approaches, emphasizing our focus on systematic evaluation under ultra-low-resource fine-tuning conditions.
> - We will explicitly discuss similarities, differences, advantages, and limitations relative to existing methods and include additional empirical comparisons where feasible.
>
> ## 2. Generality of Findings
>
> - We agree that conclusions derived from a single 0.5B model require broader validation.
> - We are conducting additional experiments on larger models and additional model families to evaluate whether the observed placement and rank-allocation trends remain consistent across architectures and scales.
> - The revised manuscript will include a discussion of the extent to which our observations generalize beyond the original experimental setup.
>
> ## 3. Validation of the Configuration Planner
>
> - We recognize that the planner was originally derived from a relatively limited experimental space.
> - To strengthen its validity, we are expanding the evaluation to include additional model sizes, datasets, and training configurations.
> - We will assess whether the planner's recommendations remain effective under these broader settings and will clearly state any limitations on its applicability.
>
> ## 4. Claims, Scope, and Limitations
>
> - We will revise the manuscript to ensure that all claims accurately reflect the scope of the experiments.
> - Conclusions will be rephrased where necessary to distinguish empirical observations from broader generalizations.
> - We will add a dedicated limitations discussion covering model scale, dataset size, architecture dependence, and training-budget considerations.
>
> ## 5. Organization and Presentation
>
> - We will improve the overall structure of the manuscript by strengthening the motivation, expanding related work, clarifying the experimental setup, reorganizing the results section, and enhancing the discussion of findings and limitations.
> - We also plan to introduce clearer summaries and presentation improvements to make the practical recommendations easier to interpret.
>
> ---
>
> We believe the planned revisions and additional analyses will substantially strengthen the paper and improve both its positioning and empirical support.

---

> ### Author Response · Authors · 2026-06-13
> **Summary of Revisions in Response to Reviewer Feedback**
>
> # Response to Reviewer f9PV
>
> We thank the Reviewer for the insightful feedback. We have substantially revised the manuscript to better position our contribution, strengthen empirical validation, and clarify the scope of our proposed configuration guidelines. The manuscript revisions have been completed, we are performing final proofreading/formatting checks, and the revised manuscript will be uploaded within the next few days.
>
> ### R3-C1: Relationship to AutoLoRA, AdaLoRA, and LoRA-drop
>
> We agree that the original manuscript did not sufficiently discuss prior work on automatic LoRA configuration. We have added a new subsection, **"Automatic LoRA Configuration,"** in Related Work that discusses:
>
> * **AdaLoRA** (Zhang et al., 2023), which dynamically allocates rank during training using importance scores.
> * **AutoLoRA** (Zhang et al., 2024), which employs meta-learning to determine adapter placement.
> * **LoRA-drop** (Zhou et al., 2024), which prunes redundant adapters during training.
>
> We clarify that our work is **complementary** rather than competitive with these approaches. While prior methods focus on adaptive configuration during training and often require additional computation, our goal is to provide a systematic empirical study of fixed LoRA configurations in ultra-low-resource settings (<200 examples) and derive practical guidance for practitioners selecting configurations before training begins.
>
> **Changes made:** Added a dedicated Related Work subsection, explicit positioning statements, and citations for AdaLoRA, AutoLoRA, and LoRA-drop.
>
> ---
>
> ### R3-C2: Need for Larger and Different Models
>
> We agree that conclusions drawn from a single 0.5B model would have limited generality. To address this concern, we conducted additional experiments using **Llama-3.2-1B-Instruct**, representing:
>
> * A different model family (Meta vs. Alibaba),
> * A larger scale (1B vs. 0.5B parameters),
> * Different pretraining and instruction-tuning procedures.
>
> Our key findings replicate across both architectures. For example, on MBPP, attention-only LoRA remains superior to combined placement on both Qwen2.5-0.5B and Llama-3.2-1B. Similarly, combined placement continues to perform strongly on GSM8K mathematical reasoning tasks.
>
> This cross-architecture consistency increases confidence that the observed trends reflect broader properties of LoRA adaptation rather than artifacts of a single model family.
>
> **Changes made:** Added cross-architecture validation results, a new Section 5.7, updates to the Abstract, Introduction, Limitations, and Conclusion, and a citation for Llama 3.2.
>
> ---
>
> ### R3-C3: Configuration Planner Derived from a Limited Experimental Space
>
> We acknowledge that our decision rules are derived from a limited experimental setting involving two tasks, two model architectures, and fixed training schedules. We therefore revised the manuscript to avoid presenting the planner as a universally applicable framework.
>
> Specifically, we now describe the proposed rules as **empirically grounded starting points** rather than universal prescriptions. They are intended to assist practitioners in selecting initial LoRA configurations and should be validated for specific applications, particularly when:
>
> * Using substantially larger models (e.g., 7B+),
> * Training on larger datasets,
> * Employing longer training schedules,
> * Applying to different task domains.
>
> We also note that the successful replication of findings on Llama-3.2-1B provides initial evidence of generalizability, while recognizing that broader validation remains future work.
>
> **Changes made:** Added caveats to the decision-rule section, expanded the Limitations discussion, reframed the practical contribution, and added future-work directions focused on larger models, additional tasks, and varied training schedules.
>
> ---
>
> ## Summary of Revisions
>
> In response to Reviewer f9PV's comments, we:
>
> 1. Added a comprehensive discussion of AdaLoRA, AutoLoRA, and LoRA-drop and clarified our relationship to prior adaptive LoRA methods.
> 2. Performed cross-architecture validation using Llama-3.2-1B-Instruct and added a new experimental section reporting these results.
> 3. Reframed the proposed configuration planner as empirically grounded guidance rather than a universal prescription and explicitly discussed its limitations and scope.
>
> We believe these revisions significantly strengthen the positioning, empirical support, and practical interpretation of our work.

---

> ### Author Response · Authors · 2026-06-21
> **Submission of Revised Manuscript**
>
> Thank you again for your valuable feedback and constructive suggestions.
>
> We have carefully revised the manuscript to address all reviewer comments and concerns. The revised version has now been uploaded to OpenReview. For ease of review, all modifications introduced in response to the reviewer feedback are highlighted in blue throughout the manuscript.
>
> Detailed point-by-point responses have been provided in our earlier comments. The revisions have substantially strengthened the paper in terms of experimental validation, clarity, reproducibility, and discussion of limitations.
>
> We sincerely appreciate the reviewers' time and effort and welcome any further comments or suggestions.